# The Mutation of the DNA-Binding Domain of Fur Protein Enhances the Pathogenicity of *Edwardsiella piscicida* via Inducing Overpowering Pyroptosis

**DOI:** 10.3390/microorganisms12010011

**Published:** 2023-12-20

**Authors:** Mimi Niu, Zhihai Sui, Guoquan Jiang, Ling Wang, Xuemei Yao, Yonghua Hu

**Affiliations:** 1State Key Laboratory of Marine Resource Utilization in South China Sea, Hainan University, Haikou 570228, China; mimi_niu@163.com; 2Key Laboratory of Biology and Genetic Resources of Tropical Crops, Institute of Tropical Bioscience and Biotechnology, Chinese Academy of Tropical Agricultural Sciences, Haikou 571101, China; 15676777363@163.com (G.J.); wangling@itbb.org.cn (L.W.); 3Key Laboratory of Biology and Genetic Resources of Tropical Crops of Hainan Province, Hainan Institute for Tropical Agricultural Resources, Haikou 571101, China; 4School of Life Sciences, Hainan University, Haikou 570228, China; 5School of Life Science, Linyi University, Linyi 276000, China; suizhihai1029@126.com; 6College of Fisheries, Huazhong Agricultural University, Wuhan 430070, China; 7Hainan Provincial Key Laboratory for Functional Components Research and Utilization of Marine Bio-Resources, Haikou 571101, China; 8School of Marine Biology and Aquaculture, Hainan University, Haikou 570228, China

**Keywords:** fish pathogen, ferric uptake regulator, N-terminal domain, stress tolerance, virulence

## Abstract

*Edwardsiella piscicida* is an important fish pathogen with a broad host that causes substantial economic losses in the aquaculture industry. Ferric uptake regulator (Fur) is a global transcriptional regulator and contains two typical domains, the DNA-binding domain and dimerization domain. In a previous study, we obtained a mutant strain of full-length *fur* of *E. piscicida*, TX01Δ*fur*, which displayed increased siderophore production and stress resistance factors and decreased pathogenicity. To further reveal the regulatory mechanism of Fur, the DNA-binding domain (N-terminal) of Fur was knocked out in this study and the mutant was named TX01Δ*fur*2. We found that TX01Δ*fur*2 displayed increased siderophore production and enhanced adversity tolerance, including a low pH, manganese, and high temperature stress, which was consistent with the phenotype of TX01Δ*fur*. Contrary to TX01Δ*fur*, whose virulence was weakened, TX01Δ*fur*2 displayed an ascended invasion of nonphagocytic cells and enhanced destruction of phagocytes via inducing overpowering or uncontrollable pyroptosis, which was confirmed by the fact that TX01Δ*fur*2 induced higher levels of cytotoxicity, IL-1β, and p10 in macrophages than TX01. More importantly, TX01Δ*fur*2 displayed an increased global virulence to the host, which was confirmed by the result that TX01Δ*fur*2 caused higher lethality outcomes for healthy tilapias than TX01. These results demonstrate that the mutation of the Fur N-terminal domain augments the resistance level against the stress and pathogenicity of *E. piscicida*, which is not dependent on the bacterial number in host cells or host tissues, although the capabilities of biofilm formation and the motility of TX01Δ*fur*2 decline. These interesting findings provide a new insight into the functional analysis of Fur concerning the regulation of virulence in *E. piscicida* and prompt us to explore the subtle regulation mechanism of Fur in the future.

## 1. Introduction

Iron plays a vital role in the growth, metabolism, and virulence of most microbial pathogens [1,2]. To combat the nutritional immunity of iron limitation in the host, the bacterial pathogens have evolved a diverse array of iron acquisition strategies [3,4]. Some pathogens, such as *Staphylococcus aureus*, *Pseudomonas aeruginosa*, and *Yersinia pseudotuberculosis*, secrete siderophores to uptake iron from most mineral and organic complexes or produce hemophores to acquire heme from hemoglobin, hemopexin, and myoglobin [5,6,7]. However, excessive intracellular iron triggers a Fenton reaction, causes the generation of highly reactive hydroxyl radicals, and damages all types of biomolecules [8]. Therefore, iron homeostasis is regulated strictly through the effective mechanism to protect the bacteria against iron-induced free radical injury [9]. In most Gram-negative and Gram-positive bacteria, the regulation of iron homeostasis is primarily mediated by the ferric uptake regulator (Fur) or diphtheria toxin repressor (DtxR) [10,11,12]. Fur is a global iron-dependent transcription factor and responds to the changed concentration of iron, which typically contains two structural domains: the N-terminal DNA-binding domain and C-terminal dimerization domain [13]. When binding with the ferrous iron under iron replete conditions, the activated Fur protein recognizes and binds the target DNA region called “Fur box” close to or in the promoter region of target genes, which represses the expression of genes involved in the iron acquisition process and other cellular metabolic processes [14]. In addition, it has been demonstrated that Fur is related to the regulation of virulence and virulence-associated factors in some pathogenic bacteria, such as *P. aeruginosa*, *Mycobacterium tuberculosis*, and *Xanthomonas vesicatoria* [15,16,17].

*Edwardsiella piscicida* (formerly called *Edwardsiella tarda*) is a Gram-negative bacterial pathogen, which infects a broad host of species, including freshwater and marine fish, animals, and humans [18]. In the aquaculture industry, *E. piscicida* causes Edwadsiellosis in many commercial fish, including channel catfish, flounder, carp, and tilapia, which leads to considerable economic losses [19]. It has been reported that the Fur protein exists in the genome of *E. piscicida* and the mutations of C92S and C95S inactivate the function of Fur, which is similar to that of *Escherichia coli* [20,21]. Our previous study constructed a functionally deficient strain of Fur (amino acids 35–109 were deleted), TX01Δ*fur*, and demonstrated that Fur regulated siderophore synthesis, the sensitivity toward reactive oxygen species (ROS), acid tolerance, and the expression of some virulence genes of *E. piscicida* [22]. It was found that Fur bound directly to the Fur box in the promoter upstream of EvpP, an effector of type-VI secretion systems (T6SSs), and consequently inhibited the function of T6SS of *E. piscicida*, which is important for the secretion of virulent factors [23]. Recently, the expression of a new virulence factor of *E. piscicida*, thioredoxin H (TrxH), was also found to be regulated by Fur [24]. Therefore, like other bacteria, the Fur protein is involved in the diverse biological processes of *E. piscicida* and has a considerable impact on the factors of physiology and infectivity [22].

In this study, we construct a different in-frame knockout mutation of the N-terminal DNA-binding domain (amino acids 8–54) of Fur in *E. piscicida* TX01 and investigate its effect on the biological properties, including the growth, siderophore production, biofilm, motility, stress tolerance, and virulence. The results provide a new insight into the specific structure–function relationship of Fur in *E. piscicida*, which allows us to thoroughly comprehend the pathogenesis of *E. piscicida*.

## 2. Materials and Methods

### 2.1. Bacteria, Cell Lines, and Fish

*E. piscicida* TX01 isolated from a diseased Japanese flounder was cultured by Luria-Bertani (LB) broth medium at 28 °C [25]. *E. coli* DH5α and S17-1λpir purchased from Tiangen (Beijing, China) and Biomedal (Sevilla, Spain), respectively, were cultured in LB medium at 37 °C. When needed, ampicillin, kanamycin, chloramphenicol, tetracycline, and polymyxin B were supplemented at the final concentrations of 100, 50, 30, 15, and 100 μg/mL, respectively. To achieve an iron depletion condition, 2,2′-dipyridyl (DP) (Sigma, St. Louis, MO, USA) was added to the medium at a final concentration of 100 μM.

Flounder gill (FG) cell, HeLa cell, mouse macrophages RAW264.7, and J774.1 cells were cultured in Dulbecco’s minimal Eagle’s medium (DMEM) (Gibco, Waltham, MA, USA) containing 10% fetal bovine serum (FBS) (Gibco, Waltham, MA, USA) in 5% CO_2_ (exclusive of FG) at 28 °C, 37 °C, 37 °C, and 37 °C, respectively. Healthy tilapias (average weight: 12.5 g) were purchased from a commercial fish farm in Haikou and maintained at ~25 °C for 2 weeks in aerated water before performing the experiments.

### 2.2. Construction of In-Frame Deletion Mutation of Fur

To distinguish the previous mutant TX01Δ*fur* [22], we named the new mutation TX01Δ*fur*2 in this study. For constructing TX01Δ*fur*2, an overlap polymerase chain reaction (PCR) was performed to delete a 141 bp fragment (residues 8 to 54) of *fur*. The primers used in this study are listed in Table 1. The upstream and downstream fragments were amplified with the primers *fur*2_up_F/*fur*2_up_R and *fur*2_down_F/*fur*2_down_R, respectively. The PCR reaction was performed in 50 μL reaction volumes containing 25 μL of a 2 × Taq PCR mix (Tiangen, Beijing, China), 2.5 μL of forward primer (10 pmol/μL), 2.5 μL of reverse primer (10 pmol/μL), 1 μL of the genome DNA of *E. piscicida* TX01, and 19 μL of double-distilled H_2_O. The PCR reaction program included an initial cycle performed at 94 °C for 10 min, followed by 30 cycles at 94 °C for 30 s, 56 °C for 30 s, 72 °C for 30–60 s, and a final elongation step of 72 °C for 10 min. After purification by gel electrophoresis, the two PCR products were fused by PCR with the primers *fur*2_up_F/*fur*2_down_R. The PCR reaction mixture and amplification program were performed as stated above. The fused PCR product was digested with B*am*H I and ligated with suicide plasmid pDM4 digested by B*gl* II to generate the recombined plasmid pDM4-Δ*fur*2, which was transformed into *E. coli* S17-1λpir using the heat shock method. *E. piscicida* TX01 was conjugated with *E. coli* S17-1λpir/pDM4-Δ*fur*2 to knock out the partial fragment of *fur* by a double-crossover homologous recombination. The positive transformant colonies were grown on an LB agar plate containing 10% sucrose and were sensitive to chloramphenicol (a marker of pDM4). The in-frame deletion mutants of *fur* were confirmed by PCR with the primer *fur*2-testF/R. This mutant strain was named TX01Δ*fur*2.

### 2.3. Growth Curve Measurement

*E. piscicida* TX01 and TX01Δ*fur*2 were cultured in the LB liquid medium for 16 h at 28 °C by shaking. After the cultures were washed with PBS, the pellets were diluted to the same cell density of about 10^7^ colony-forming units per milliliter (CFU/mL) with PBS. Then, 50 μL of the abovementioned bacterial suspensions were transferred to 5 mL fresh LB medium with or without DP and incubated at 28 °C for a continuous 26 h. To monitor the growth trends, the samples were withdrawn aseptically at an interval of 2 h, and the optical density (OD) values were measured at 600 nm with an ultraviolet and visible (UV-Vis) spectrophotometer. The experiment was repeated three times.

### 2.4. Detection of Siderophore Production

A chromoazurol S (CAS) agar assay was used to detect the siderophore production, which changed the color from blue to orange/yellow [26]. A total of 10 μL of TX01 and TX01Δ*fur*2 (OD_600_ = 1.0) were spotted onto a chromoazurol S (CAS) agar plate. After incubation for 24–72 h at 28 °C, a yellow–orange halo appearing around the bacterial colony was interpreted as the production of siderophores. 

### 2.5. Biofilm Formation Assay

The overnight cultures of TX01 and TX01Δ*fur*2 were diluted to 10^5^ CFU/mL. Then, 200 μL of the diluted cultures were transferred into a 96-well polystyrene microtiter plate (Nunc, Denmark) and incubated at 28 °C for 24 h under static conditions. After washing this twice with sterile phosphate-buffered saline (PBS, pH 7.2) and fixation by Bouin’s reagent for 1 h, the samples were stained with a 2% crystal violet solution for 20 min. After being washed with running water and air dried, the dye bound to the cells was resolubilized with ethanol and the OD_570_ was measured.

### 2.6. Swimming Motility Assays

The overnight cultures of TX01 and TX01Δ*fur*2 were transferred to the fresh LB liquid medium and cultured to an optical density (OD_600_) of 0.5. A total of 10 μL bacterial suspensions was dotted on the center of the LB plates with 0.3% (*w*/*v*) agar. Then, after incubation at 28 °C for 24 h, swimming motility was assessed and the diameters of the swimming halos were measured.

### 2.7. Resistance to Environmental Stress (Acid, Manganese, and Temperature)

TX01, TX01∆*fur*, and TX01∆*fur*2 were cultured in LB medium to achieve an exponential phase. To determine acid resistance, the bacteria was streaked on the LB agar plates at a pH = 5 and incubated for 24 h at 28 °C. For the quantitative analysis, the 3 strains were adjusted to a cell density of 10^6^ CFU/mL and incubated in the LB medium with different pHs (pH = 4.0, 5.0, and 7.0) at 28 °C for 4 h. Then, the populations of cultivated bacteria were enumerated by dilution plate counts and the relative survival rate was calculated as the ratio of bacterial counts at pHs 4.0 or 5.0 to that at pH = 7.0.

The assay of manganese resistance was performed as previously reported [22]. A total of 10 μL of TX01, TX01Δ*fur*, and TX01Δ*fur*2 was streaked on the LB agar plates supplemented with or without 7 mM of MnCl_2_. The plates were incubated at 28 °C for 24 h and the bacterial growth was examined.

The assay of temperature tolerance was performed as previously reported [22]. TX01, TX01Δ*fur*, and TX01Δ*fur*2 were adjusted to the cell density of 10^6^ CFU/mL and incubated in the LB medium at 28 °C or 42 °C for 4 h. After washing with PBS and performing a resuspension with PBS, the total bacterial populations were counted using the plate-dilution method and the relative survival rate was calculated as the ratio of bacterial counts at 42 °C to that at 28 °C.

### 2.8. The Flagella Observation by Transmission Electron Microscopy (TEM)

TX01 and TX01∆*fur*2 were cultured in LB solid medium. The bacteria were resuspended with PBS. Then, the suspensions were dropped onto a copper grid and negatively stained with 1% phosphotungstic acid. Electron microscopy was performed with a HT7700 transmission electron microscope (Hitachi, Tokyo, Japan).

### 2.9. Bacterial Invasion of Eukaryotic Cell Lines

To detect the bacterial invasion of nonphagocytic cells, FG and HeLa cells were seeded and grown to a 100% confluence in a 24-well plate at 28 °C and 37 °C, respectively. Logarithmic TX01 and TX01∆*fur*2 were resuspended in PBS and adjusted to the cell density of 10^7^ CFU/mL. The bacteria were added to each well at a multiplicity of infection (MOI) ratio of 20:1, and the plates were centrifuged at 800× *g* for 5 min. After being incubated for 2, 6, and 10 h at 28 °C, the cells were washed with PBS 3 times and lysed with 500 μL of 1% Triton X-100 for 10 min. Then, series diluted suspensions were plated onto LB agar plates supplemented with 30 µg/mL of tetracycline and the colony-forming units (CFUs) on the plates were counted after incubation at 28 °C for 24 h.

To detect the intracellular replication of the phagocytic cells, RAW264.7 and J774.1 cells with 100% confluence rates were infected with TX01 and TX01∆*fur*2 at an MOI of 50:1 in a 24-well plate, respectively. After incubation at 28 °C for 2 h, the cells were washed with PBS, and a fresh DMEM containing 200 µg/mL of gentamicin was added for 2 h to kill extracellular bacteria. After being washed 3 times with PBS, the cells were cultured in DMEM with 10 µg/mL of gentamicin at 28 °C for 0, 2, 4, 6, 8, and 10 h. At different time points, the culture medium was collected and the infected cells were lysed with 500 µL of 1% Triton X-100 for the number of viable extra- and intracellular bacteria. Serial dilutions of the supernatant and cellular lysates were plated on the LB agar plates, and the number of bacteria was quantified by incubation at 28 °C for 24 h.

### 2.10. Fluorescence Microscopy

TX01 and TX01∆*fur*2 were transformed with the plasmid pmCherry-N1-encoded mCherry protein with red fluorescence by electroporation with Gemini Twin Waveform Electroporation systems (Harvard Apparatus BTX, Holliston, MA, USA). As described above, the FG, HeLa, RAW264.7, and J774.1 cells were infected with mCherry-labeled TX01 and TX01∆*fur*2 in 35 mm confocal dishes at 28 °C for different amounts of time. After being washed 3 times with PBS and performing fixations with 4% PFA for 20 min, the cells were washed as above and stained with DAPI for 10 min at room temperature. The cells were again washed as above and observed with a confocal laser scanning microscope (Olympus Fluoview FV1000, Tokyo, Japan).

### 2.11. Pyroptosis Analysis

TX01 and TX01∆*fur*2 infected the J774.1 cells at an MOI of 50:1 at 28 °C. After incubation for 6 h, the morphology of pyrolytic cells was observed under a light microscope. For caspase-1 detection, the plates were centrifuged at 5000× *g* for 5 min and washed with PBS 3 times. Then, the cells were lysed in a radio immunoprecipitation assay (RIPA) lysis buffer supplemented with protease inhibitor phenylmethylsulfonyl fluoride (PMSF) (1 mM) for 5 min. After centrifugation at 12,000× *g* for 5 min, the supernatants were resolved by 10% sodium dodecyl sulfate-polyacrylamide gel electrophoresis (SDS-PAGE) and electroblotted onto polyvinylidene fluoride (PVDF) membranes. The membranes were blocked with 5% BSA in PBST (PBS containing 0.1% Tween 20) and then incubated with rabbit anti-pro-caspase-1 + p10 + p12 (ab179515, abcam, Boston, MA, USA) or anti-β-actin (ab8227, abcam, Boston, MA, USA) (as an internal control) primary antibodies (1:2000) overnight at 4 °C. After being washed with PBST 3 times, the samples were incubated with horseradish peroxidase (HRP)-conjugated goat anti-rabbit-IgG secondary antibody (1:1000) for 90 min at room temperature. After being washed as mentioned above, the protein bands were visualized by using an enhanced chemiluminescence system reagent (P0018S, Beyotime Biotech, Shanghai, China).

### 2.12. Cytotoxicity Assay and IL-1β Detection

TX01 and TX01∆*fur*2 infected the J774.1 cells at an MOI of 50:1 at 28 °C. After being incubated for 4, 6, and 8 h, the plates were centrifuged at 5000 rpm for 10 min and the supernatants were transferred to a 96-well plate. For the cytotoxicity assay, the detection of lactate dehydrogenase (LDH) release was determined by using LDH assay kits (Beyotime Biotech, Shanghai, China) according to the manufacturer’s instructions. A total of 60 μL of an LDH reaction solution was added to each well and incubated for 30 min at room temperature. The absorbance at 490 nm was determined using a microplate reader. The cells were lysed with 1% Triton-X-100 as the maximum LDH. Percentage cytotoxicity was expressed as (sample LDH-background LDH)/(maximum LDH-background LDH) × 100%. For IL-β detection, the expression level of IL-β in the supernatants was measured using mouse IL-1β ELISA kits (Beyotime Biotech, Shanghai, China) according to the manufacturer’s instructions. 

### 2.13. Experimental Challenge of Bacterial Disseminations In Vivo

Logarithmic TX01 and TX01Δ*fur*2 were resuspended in PBS to 10^6^, 10^7^, 10^8^, and 10^9^ CFU/mL. Healthy tilapias were randomly divided into 9 groups (n = 30/group) and infected with 100 µL of different diluted bacterial suspensions or PBS (as control) by intramuscular injections. Then, the infected fish were monitored daily for mortality for 15 days. For the tissue dissemination analysis, logarithmic TX01 and TX01Δ*fur*2 were resuspended in PBS to 10^8^ CFU/mL. Healthy tilapias were infected with 100 µL of bacterial suspension. The spleen and kidney tissues were collected from the 5 fish euthanized with tricaine methanesulfonate (MS-222) under sterile conditions at 24 and 48 h post-infection. The mixed tissues were homogenized in PBS using an electrically driven tissue homogenizer (TGrinder, Tiangen, China) and bacteria recoveries were analyzed by diluted plate counting. The experiment was repeated three times.

### 2.14. Reactive Oxygen Species (ROS) Production

The generation of ROS was estimated with the 2′,7′-dichlorofluorescein diacetate (DCFH-DA) assay. Tilapias’ head kidney macrophages (HKMs) were extracted, as described previously [27]. The HKMs were infected with TX01, TX01Δ*fur*2, and PBS (as the control) at an MOI of 10:1 for 2 h at 28 °C. Then, the L-15 medium containing 10 μM of DCFH-DA was added to the samples for cultivation at 28 °C for 30 min. After being washed with PBS, the cells were lysed with 1% Triton-X-100 for 15 min and centrifuged at 5000 rpm for 5 min. The fluorescence intensity of the supernatant was measured in a fluorescence spectrophotometer at an excitation/emission wavelength of 488/525 nm. The HKMs were treated with 100 μM of hydrogen peroxide as the positive control. The relative fluorescence intensity was expressed as [(sample − blank control)/(positive control − blank control)] × 100%

### 2.15. Statistical Analysis

All experiments were performed at least three times, and the statistical analyses were performed with SPSS 23 software (SPSS Inc., Chicago, IL, USA). The data were analyzed using an analysis of variance (ANOVA) and expressed as the mean ± standard error of the mean (SEM) (N = 3). N, the number of times the experiment was performed. The statistical significance was defined as *p* < 0.05.

## 3. Results

To further explore the regulatory mechanism of Fur and to evaluate Fur’s importance to the virulence of *E. piscicida*, the DNA-binding domain of Fur (N-terminal, amino acid residues 8 to 54) was knocked out using the in-frame deletion system, and the obtained mutant was named TX01Δ*fur*2. Then, the effects of the deletion of the Fur N-terminal domain on the physiology and infectivity were examined and analyzed.

### 3.1. Effect on Growth and Siderophore Production

The growth curves showed that, when cultured in the LB medium, TX01Δ*fur*2 displayed a slower exponential growth rate and a lower maximum cell density in the stationary phase than the wild-type strain TX01 (Figure 1A). When cultured in the iron-restricted environment created by adding 100 µM of DP, the growth rates of the two strains were considerably retarded, but the mutant grew slightly after 16 h (Figure 1A). The CAS plate assay showed that the color change in the CAS-positive reaction from blue to yellow was observed around TX01Δ*fur*2 and TX01Δ*fur* (Figure 1B), which indicated that the two strains exhibited a good ability for siderophore production. However, only a slightly yellow color emerged around TX01. These results demonstrate that whole Fur or N-terminal deactivations lead to the enhancement of siderophore production/accumulation.

### 3.2. Effect on Resistance against Environmental Stress

The sensitivity to environmental stress of TX01, TX01Δ*fur*, and TX01Δ*fur*2 was measured by plate streaking and dilution coating methods. When exposed to acid stress (pH 5.0), the growth rates of TX01Δ*fur*2 and TX01Δ*fur* were similar, and both were much better than that of TX01 (Figure 2A). Consistently, the survival rates at pHs 4.0 and 5.0 of TX01Δ*fur*2 were 22.11% and 81.57%, respectively, which were significantly higher than that of TX01 (7.44% and 40.60%, respectively) (Figure 2B), and the survival rate of TX01Δ*fur*2 was also higher than TX01Δ*fur* (Appendix A). Similarly, like TX01Δ*fur*, TX01Δ*fur*2 grew normally on the LB agar plate with 7 mM of MnCl_2_ (Figure 2C), but TX01 hardly grew under the same condition [22]. When exposed to a high temperature (42 °C), the survival rate of TX01Δ*fur*2 was 59.09%, which was significantly higher than that of TX01 (21.45%) and TX01Δ*fur* (42.55%) (Figure 2D and Appendix A, respectively). These results indicate that, similar to TX01Δ*fur*, the capability of TX01Δ*fur*2 to resist adversity, such as acid stress and high temperatures, is stronger than that of TX01.

### 3.3. Effect on the Abilities of Biofilm Formation and Motility

The biofilm formation ability of *E. piscicida* was investigated and quantified by the crystal violet assay. The results show that, compared with TX01, the ability of TX01∆*fur*2 to adhere to the well surface of a polystyrene plate is reduced and the biofilm production of TX01∆*fur*2 is significantly lower (1.6-fold less) (Figure 3A).

The bacterial mobility was estimated by detecting bacterial swimming activity on the LB agar plate. The results show that, compared with TX01, the motility of TX01∆*fur*2 is significantly inhibited after incubation for 24 and 48 h (upper section in Figure 3B). The swimming zone diameters of TX01∆*fur*2 were 2.03 ± 0.25 and 8.50 ± 1.50 mm at 24 and 48 h, respectively, which were significantly smaller than those of TX01 (18.00 ± 2.00 and 27.17 ± 1.04 mm) (lower section in Figure 3B). To explore whether decreased motility was related with flagellum formation, flagellar morphology was performed by TEM. The result shows that the number of flagella around TX01∆*fur*2 is not significantly different to TX01 (Appendix A).

### 3.4. Effects on Lethality and Infectivity In Vivo

Furthermore, we wanted to know whether the deletion of the Fur N-terminal domain was related to bacterial pathogenicity. Healthy tilapias were infected with the same density gradients of TX01 and TX01∆*fur*2, and their mortalities were monitored. At the end of the monitored period (15 days), dead fish were counted and the result showed that, although the survival rates of fish infected with TX01 and TX01∆*fur*2 at a cell density of 10^8^ CFU/fish were the same (0%), the survival rate of TX01∆*fur*2-infected fish was significantly lower than that of TX01-infected fish at the cell densities of 10^5^, 10^6^, and 10^7^ CFU/fish (Figure 4A). Similar results were observed in two of the other repeated experiments. At the same time, the tissue disseminations of TX01 and TX01∆*fur*2 in the spleen and kidneys were determined at 24 and 48 hpi, and the results showed that bacterial recoveries from TX01∆*fur*2-infected fish presented no significant difference to those from TX01-infected fish at 24 and 48 hpi (Figure 4B). To explore the reason for the tilapias’ high mortality rate caused by TX01∆*fur*2, ROS production levels in the macrophages were examined. The result shows that the ROS levels generated by the HKMs of tilapias infected with TX01Δ*fur*2 are significantly lower than those in the TX01-infected cells (Figure 4C). These results indicate that the TX01∆*fur*2 proliferation level in host tissues is equivalent to TX01, but the lethal effect of TX01∆*fur*2 on the host is obviously greater than that of TX01, and the reason for the increased virulence of TX01∆*fur*2 is perhaps a stronger immune-escape response, such as the inhibition of ROS by TX01∆*fur*2.

### 3.5. Effects on Invasion and Replication In Vitro

The results indicate that the deletion of the N-terminal domain of Fur enhances the virulence of *E. piscicida*, which is different from the previously obtained result that shows that whole Fur deficiency greatly decreases bacterial virulence [22]. Since *E. piscicida* possesses the capacity to adhere to, invade, and replicate in the host cells [28], we assessed the virulence of TX01Δ*fur*2 using nonphagocytic (Hela and FG) and phagocytic (RAW264.7 and J774.1) cells. For the experiments of the adhesion to and invasion of nonphagocytic cells, the results show that the quantity of TX01Δ*fur*2 obtained from HeLa cells, though similar to that of TX01 at 2 hpi, is significantly higher than TX01 at 6 and 10 hpi (Figure 5A), which is confirmed by a confocal microscope observation (Figure 5B). A similar result was observed for the infected FG cells. These results demonstrate that the deletion of the N-terminal domain of Fur enhances the adhesion to and invasion of *E. piscicida* related to nonphagocytic cells.

For the intracellular replication of phagocytic cells, the results show that the intracellular quantity of TX01Δ*fur*2 collected from RAW264.7 cells, although not different from that of TX01 at 2 and 4 hpi, is significantly lower than TX01 at 6 and 8 hpi (Figure 6A), which is confirmed by the confocal microscope observation (Figure 6B). However, the result contradicts that of nonphagocytic cells. Given the result of the in vivo experiment, we examined the bacterial number in the culture medium and the result showed that the quantity of TX01Δ*fur*2 obtained from the supernatant of infected RAW264.7 cells was significantly higher than that of TX01 at 6 and 8 hpi (Figure 6C); therefore, the total amount of TX01Δ*fur*2 was comparative to that of TX01 at all examined time points, except 6 hpi (Figure 6D). A similar result was obtained using J774.1 cells. These findings demonstrate that, compared to TX01, TX01Δ*fur*2 possesses a stronger capability to damage phagocytes and is easier to release from cells, which indicates its enhanced virulence.

### 3.6. Effect on the Inflammatory Responses of Macrophage Cells

To explore what occurs in macrophages infected with TX01∆*fur*2, we investigated the inflammation reaction using J774.1 cells. Firstly, the cytotoxicity caused by bacterial infection was examined and the result showed that LDH release induced by TX01∆*fur*2 infection was significantly greater than that of TX01 at 4, 6, and 8 hpi (Figure 7A). A similar result was observed when an IL-1β release level was detected (Figure 7B). Furthermore, more obvious pyroptotic phenotypes, such as cell swelling and membrane rupture, were observed in TX01∆*fur*2-infected cells, compared with TX01-infected cells (Figure 7C). Consistently, we found that cleaved product p10 of the caspase-1 activation generated during TX01∆*fur*2 infection was stronger than that of TX01 infection (Figure 7D). These results show that the deficiency of the Fur N-terminal domain enhances the capability of *E. piscicida* to evoke caspase-1-dependent pyroptosis.

## 4. Discussion

As a global transcriptional regulator in many pathogenic bacteria, Fur regulates a wide spectrum of genes involved in iron utilization, metabolism, and pathogenicity [15,29]. For *E. piscicida* TX01, the *fur* gene was confirmed to be present in the genome, which possessed two typical domains, like the Fur proteins of *E. coli* and *E. piscicida* ATCC 23685 [20,30]. Also, our previous study demonstrated that the deletion of two domains of Fur (35 to 108), named TX01Δ*fur*, had significant influences on the properties of *E. piscicida*, including increased siderophore production and decreased pathogenicity [22]. Some reports have suggested that the N-terminal helix of Fur is required for efficient DNA binding at the Fur box and for gene regulation [31]. In this study, we constructed a new mutant, *E. piscicida* TX01Δ*fur*2, where the N-terminal DNA-binding domain of Fur (residues 8 to 54) was deleted, and obtained some interesting results.

Firstly, we observed that the growth ability of TX01Δ*fur*2 was slightly impaired in the LB medium, which was similar to the results for TX01Δ*fur* and *fur* mutants in other bacteria, such as *A. salmonicida*, *Staphylococcus aureus*, and *Clostridium acetobutylicum* [22,32,33,34]. We speculated that lacking a negative regulation of Fur produced excess intracellular iron, which promoted the formation of reactive oxygen species (ROS) via the Fenton reaction to cause bacterial damage and decrease its growth [8]. This conjecture was affirmed by the results that siderophore production levels in TX01Δ*fur*2 and TX01Δ*fur* were considerably higher than TX01. Fur negatively regulates siderophore biosynthesis in many bacteria, such as *E. coli* and *P. aeruginosa* [35,36], and a CAS plate analysis is a creditable method to detect siderophores and used for various bacteria, such as *P. fluorescens* and *Legionella pneumophila* [37,38]. These findings demonstrate that the deficiency of the Fur N-terminal domain leads to the damage of the regulation function related to iron absorption, which is consistent with the result for TX01Δ*fur*. The mutation of Fur induced the upregulated expression of the genes for siderophore production, iron-related transporters, and receptor proteins in many bacteria, which was also observed in TX01Δ*fur* by isobaric tags for relative and absolute quantitation (iTRAQ) analyses [22,39,40]. 

The Fur protein is involved in mediating the manganese-dependent regulation of gene expression, and manganese imitates iron to combine with Fur; so, mutants of *fur* frequently are highly resistant to manganese [41,42]. Consistently, the ability of TX01Δ*fur*2 to resist manganese stress is comparative to that of TX01Δ*fur*, which is significantly stronger than TX01 [22]. Since manganese is crucial for acid tolerance in several bacterial species [43,44], the acid tolerance of *E. piscicida* is also examined in this study. We found that TX01Δ*fur*2 exhibited much stronger acid tolerance levels than TX01 and TX01Δ*fur*. However, the Fur mutants in other bacteria, such as *Helicobacter pylori*, are more sensitive to low pH levels [45]. Perhaps, Fur in *E. piscicida* negatively regulates the expression of manganese-related genes, like *Yersinia pestis* [46], or the acidic resistance system may be negatively regulated by Fur.

Accumulating evidence suggests that Fur is closely associated with bacterial pathogenicity. In many pathogens, including *E. ictalurid*, *A. salmonicida*, *P. fluorescens*, and *Acidovorax*. *Citrullin* [30,32,37,47], the *fur* knockout reduces bacterial virulence. In our previous study, the *fur* mutation (TX01Δ*fur*) also impaired the pathogenicity of *E. piscicida* [22]. Contrary to our expectations, in this study, the deletion of the Fur N-terminal domain (TX01Δ*fur*2) remarkably enhanced the virulence of *E. piscicida*, including a higher lethality rate for healthy tilapias, a stronger adhesion to and invasion of nonphagocytic cells, and greater damage to phagocytes, although the capability of biofilm formation and the motility of TX01Δ*fur*2 were abated. Moreover, the increased virulence was not dependent on the bacterial number. We immediately analyzed the reasons for the increased virulence levels. Our results show that TX01Δ*fur*2 strongly suppresses the ROS production of HKM. It was confirmed that the inhibition of ROS contributed to survival in fish phagocytes [48]. When we infected nonphagocytic cells, more TX01Δ*fur*2 than TX01 adhered to and invades the cells. When we infected phagocytes, less TX01Δ*fur*2 than TX01 survived in the cells, but more TX01Δ*fur*2 than TX01 was released into the supernatant, which indicated that the cells were damaged. Consistently, more obvious pyroptotic phenotypes, including cell swelling and membrane rupture, were observed in TX01∆*fur*2-infected than TX01-infected cells, which was confirmed by the fact that TX01Δ*fur*2 induced higher levels of cytotoxicity, IL-1β, and p10 in the macrophages than TX01. It was demonstrated that viable *E. piscicida* released from the macrophages via pyroptosis enhanced the infectivity of epithelial cells and promoted the pathogenicity of the host [49]. Also, we been reported that *E. piscicida* prevented NLPR3 inflammasome activation-mediated pyroptosis via the T6SS effector EvpP, which was regulated by Fur [23,50]. These results indicate that the knockout of the DNA-binding domain of Fur increases the pathogenicity of *E. piscicida*. However, a further exploration is needed to investigate how Fur with only the C-terminus domain affects the expression levels of virulence genes.

## 5. Conclusions

In this study, we constructed a knockout of the DNA-binding domain of Fur and characterized its effects on the virulence-related functions in *E. piscicida* TX01. Although the mutation of the Fur N-terminal domain impaired bacterial growth, it increased siderophore production levels and the resistance to environmental stress, including low pH, manganese, and high temperature. More importantly, the mutation boosted the ability of *E. piscicida* to adhere to and invade nonphagocytic cells, enhanced the ability of *E. piscicida* to destroy phagocytes via overpowering pyroptosis, and further increased the global virulence of *E. piscicida* in the host. These interesting findings provide new insights into the functional analysis of Fur concerning the regulation of virulence in *E. piscicida* and prompt us to explore the subtle regulation mechanism of Fur in the future.

## Figures and Tables

**Figure 1 microorganisms-12-00011-f001:**
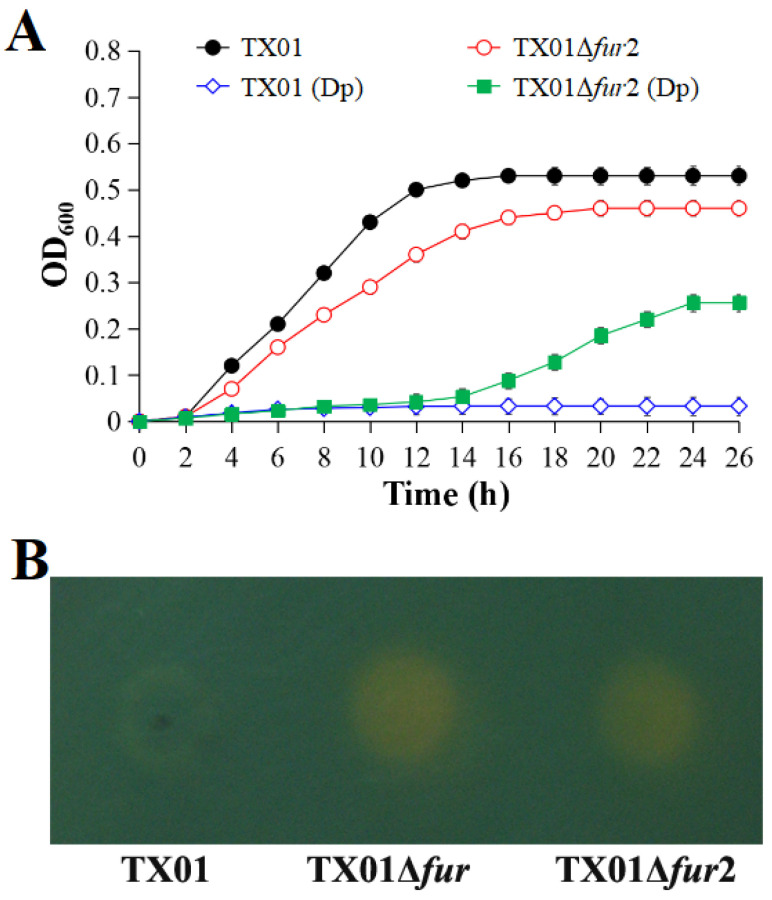
Growth analysis and siderophore production of *Edwardsiella piscicida*. (**A**) TX01 and TX01Δ*fur*2 were cultured in LB liquid medium supplemented with or without 100 µM of 2,2′-dipyridyl (DP) at 28 °C for 26 h. The absorbance at OD_600_ of the cell density was measured at the interval of 2 h. Data are presented as means ± SEMs (N = 3). N: the number of independent experiments. (**B**) Logarithmic TX01, TX01Δ*fur*, and TX01Δ*fur*2 (OD_600_ = 1.0) were spotted onto CAS agar plates and incubated at 28 °C for 24–72 h. The color change from blue to yellow around bacterial colony indicates siderophore production.

**Figure 2 microorganisms-12-00011-f002:**
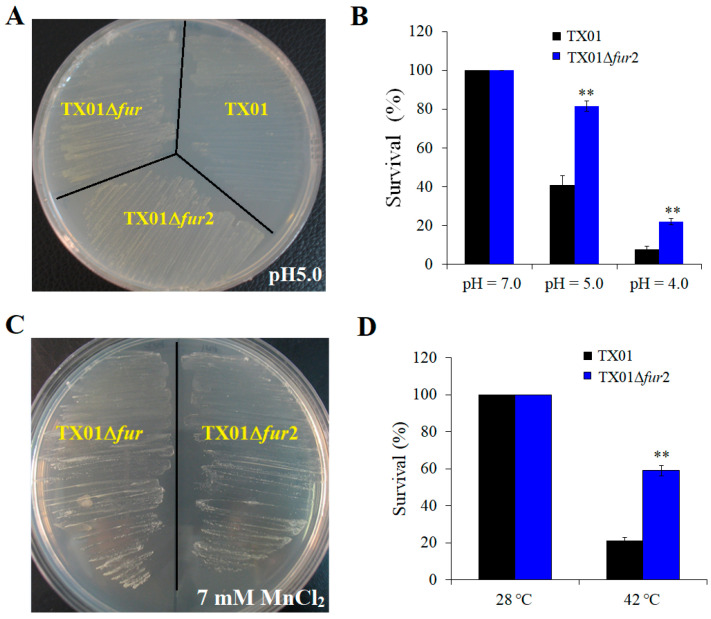
Resistance of *Edwardsiella piscicida* to environmental stress. (**A**) TX01, TX01∆*fur*, and TX01∆*fur*2 in the logarithmic phase were streaked onto the LB agar plate with a pH = 5.0 and incubated at 28 °C for 24 h. (**B**) TX01 and TX01∆*fur*2 were incubated in LB medium at pHs 4.0, 5.0, and 7.0 for 4 h. The populations of surviving bacteria were counted by the plate-dilution method and the survival rates were calculated. (**C**) TX01∆*fur*2 and TX01∆*fur* were streaked onto LB agar plates with or without (control) 7 mM of MnCl_2_, and the plates were incubated at 28 °C for 48 h. (**D**) TX01 and TX01∆*fur*2 were incubated in LB medium for 4 h at 28 °C and 42 °C, respectively. After washing, the populations of surviving bacteria were counted in diluted plated and the survival rate was calculated. Data are the means of three independent experiments and presented as mean ± SEM. **, *p* < 0.01.

**Figure 3 microorganisms-12-00011-f003:**
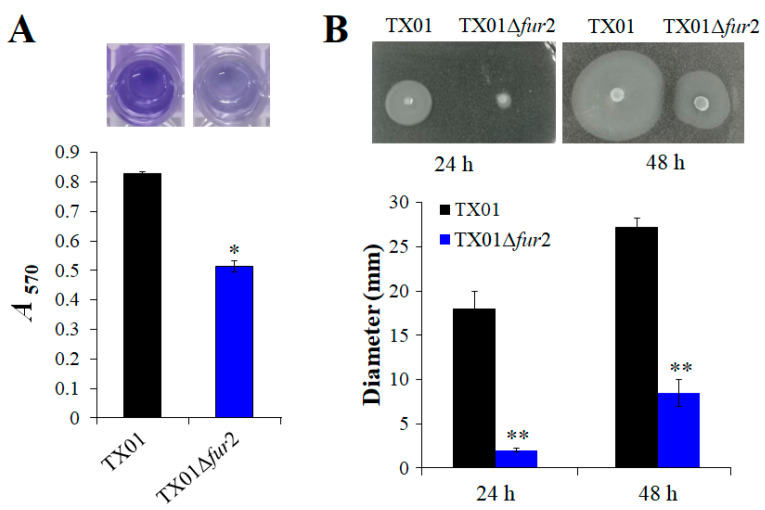
The capabilities of bacterial biofilm formation and swimming. (**A**) The biofilm formation assay. TX01 and TX01∆*fur*2 (10^5^ CFU/mL) were incubated on a 96-well polystyrene microtiter plate at 28 °C for 24 h. After washing, the wells were stained with a crystal violet solution. The dye bound to the cells was resolubilized by ethanol and OD_570_ was measured. (**B**) The plate-based assay for swimming motility. A total of 10 μL of logarithmic TX01 and TX01∆*fur*2 was spotted onto the center of the LB plates with 0.3% (*w*/*v*) agar. The plates were incubated at 28 °C, and the motility zone diameter was measured. Data are the means of three independent assays and presented as means ± SEMs. *, *p* < 0.05; **, *p* < 0.01.

**Figure 4 microorganisms-12-00011-f004:**
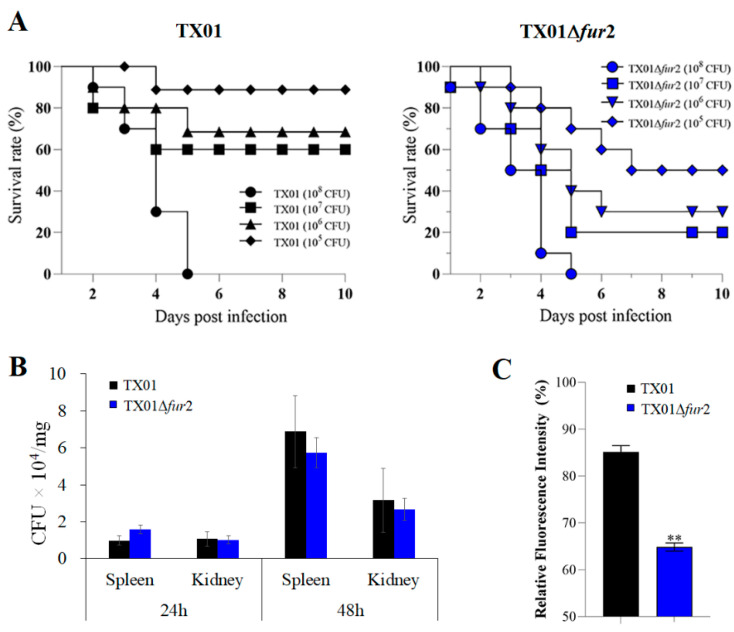
In vivo infectivity of *Edwardsiella piscicida* in tilapia. (**A**) Survival of tilapia infected with *E. piscicida*. Healthy tilapias were infected with 100 µL of bacterial suspensions of TX01 and TX01Δ *fur*2 (10^6^, 10^7^, 10^8^, and 10^9^ CFU/mL) and accumulated mortality was monitored for 15 days (only 10 days are shown since no more deaths occurred after 10 days). (**B**) Bacterial dissemination in the tissues. Tilapias were infected with 100 µL of bacterial suspensions of TX01 and TX01Δ *fur*2 (10^8^ CFU/mL). At 24 and 48 hpi, bacterial recoveries from the spleen and kidney tissues were determined. (**C**) The detection of ROS production of fish macrophages. HKMs of tilapias were extracted and infected with TX01 and TX01Δ *fur*2; the ROS level was detected by a fluorescence spectrophotometer. Data from three independent experiments are presented as means ± SEMs. **, *p* < 0.01.

**Figure 5 microorganisms-12-00011-f005:**
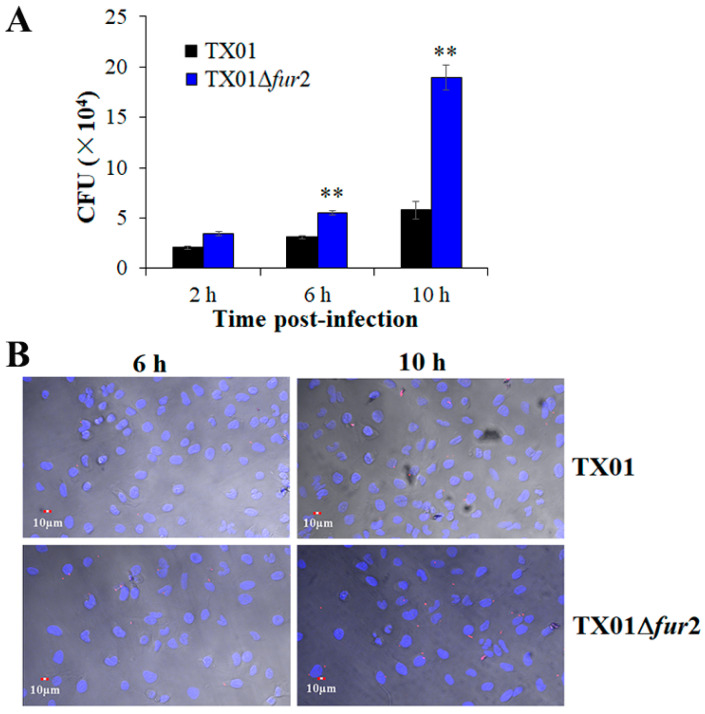
The invasion of *Edwardsiella piscicida* in nonphagocytic cells. (**A**) HeLa cells were infected with TX01 and TX01∆*fur*2 for 2, 6, and 10 h. After washing with PBS at different time points, the cells were lysed with Triton-X-100 and the numbers of bacteria were determined. (**B**) HeLa cells were infected with mCherry-labeled TX01 and TX01∆*fur*2 for 6 and 10 h, respectively. After washing and fixation were performed, the cells were stained with DAPI and observed with a confocal laser scanning microscope (CLSM). Data are the means of three independent experiments and presented as means ± SEMs (N = 3). N: the number of times the experiment was performed. **, *p* < 0.01.

**Figure 6 microorganisms-12-00011-f006:**
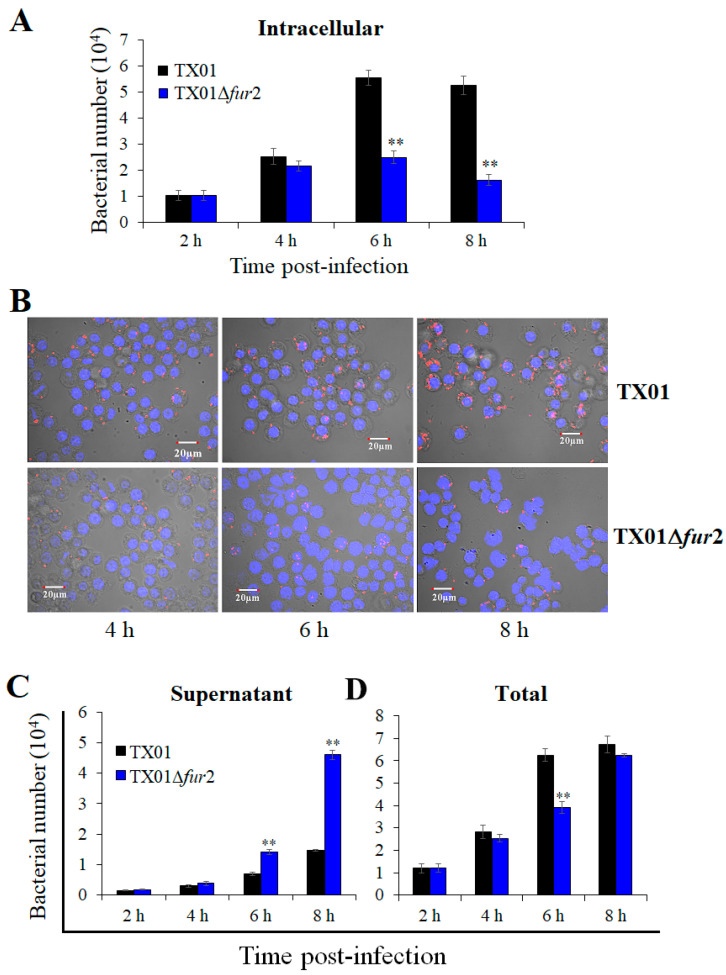
The survival of *Edwardsiella piscicida* in phagocytic cells. RAW264.7 cells were infected with TX01 and TX01∆*fur*2 for 2 h. After extracellular bacteria were washed and killed, the cells were incubated further for 2, 4, 6, and 8 h. The culture medium was absorbed and supernatant bacteria were determined (**C**). Then, the cells were lysated and intracellular bacteria were determined (**A**). (**B**) RAW264.7 cells were infected with mCherry-labeled TX01 and TX01∆*fur*2, as previously mentioned. Following fixation, the cells were stained with DAPI and observed with a confocal laser scanning microscope (CLSM). (**D**) The sum of (**A**,**C**). Data are the means of three independent experiments and presented as means ± SEMs (N = 3). N: the number of times the experiment was performed. **, *p* < 0.01.

**Figure 7 microorganisms-12-00011-f007:**
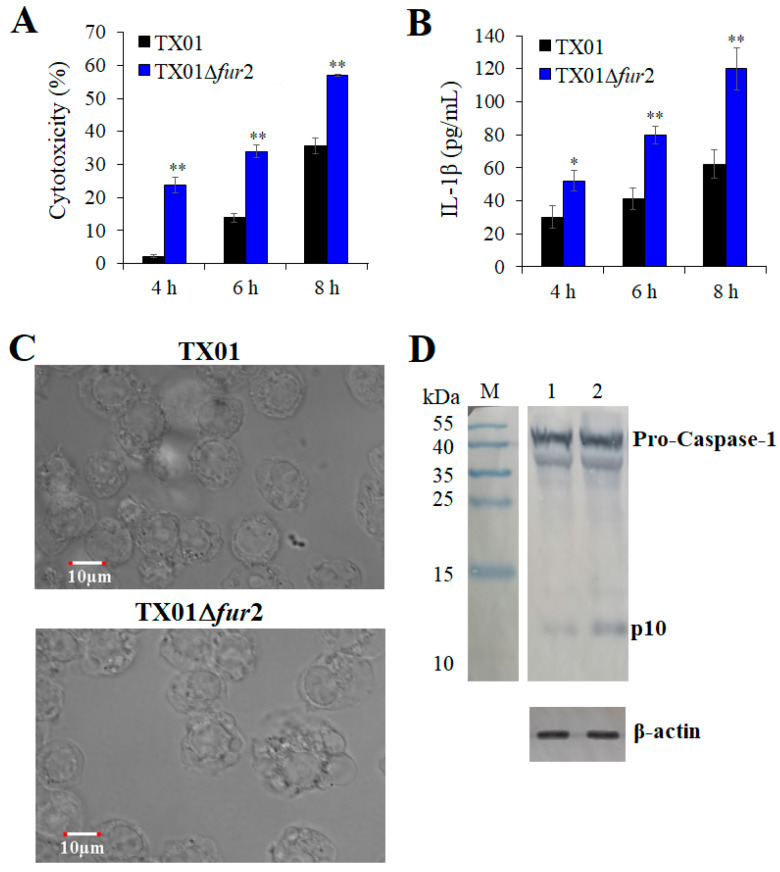
The detection of the inflammation reaction induced by *Edwardsiella piscicida*. J774.1 cells were infected with TX01 and TX01∆*fur*2 for various times. After centrifugation, the supernatant was used to detect the release of LDH (**A**) and IL-1β (**B**). (**C**) The cells infected for 6 h were observed by confocal laser scanning microscopy (CLSM). (**D**) The expressions of pro-caspase-1 and p10 were detected by Western blots. Line 1, TX01-infected J774.1 cells; line 2, TX01∆*fur*2-infected J774.1 cells. Data are presented as means ± SEMs (N = 3). N, the number of times the experiment was performed. *, *p* < 0.05; **, *p* < 0.01.

**Table 1 microorganisms-12-00011-t001:** Oligonucleotide primers used in this study.

Name	Sequence (5′→3′) ^#^
*fur*2-up-F	AGCGGGATCCGACGCCTGGGTCAAGCAGC
*fur*2-up-R	GTATACCGCGGTGTTGTTGTCAGTCA
*fur*2-down-F	ACAACAACACCGCGGTATACCGCGTGCTCAACCAG
*fur*2-down-R	AGCGGGATCCTTAGGCCTTTTCGTCGTGCA
*fur*2-test-F	AGGACAGAATCCGAATGAC
*fur*2-test-R	TTCACGCTGACGCTTCT

^#^ Underlined nucleotides are restriction sites for the B*amH* I enzyme.

## Data Availability

The data in this study are readily available upon reasonable request from the corresponding authors.

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
