# Peer review of "The Mutation of the DNA-Binding Domain of Fur Protein Enhances the Pathogenicity of Edwardsiella piscicida via Inducing Overpowering Pyroptosis"

_microorganisms, 2023, doi:10.3390/microorganisms12010011_

Round 1

Reviewer 1 Report

Comments and Suggestions for Authors

The study by Niu and colleagues reports the construction of a mutant of Edwardsiella piscicida (in-frame deletion mutation in the Fur DNA-binding domain) and its characterization in relation to growth kinetics, siderophore production, motility, ability to form biofilm on abiotic surface, resistance to environmental stressors, adhesion and invasion in eukaryotic cells in vitro and in vivo infection in tilapia.

 Some issues should be clarified, mainly related to methodology. A clear description of the methodology for interpreting the results is crucial.

- Was the deletion in the DNA-binding domain in the mutant strain confirmed only by PCR?

-Why was the TX01∆fur mutant strain included in only a few experiments?

-The standardization of bacterial inocula for experiments must be clearly informed since the authors carry out a comparative analysis between wild-type and mutant strains. For example:  How did the authors ensure that the initial inoculum for growth curve analysis contained the same cell density for the wild-type and mutant strains? According to the methodology, 50 uL of the 16 h cultures were added.

-Transmission electron microscopy: The purpose and results of this methodology are not shown in the main body of the manuscript, only in the supplementary material.

-2.9. Does the assay describe bacterial adhesion or invasion into eukaryotic cells?

-2.10. How were the wild-type and mutant strains transformed? Which eukaryotic cells were subjected to the invasion assays? On what equipment were the microscopic analyses carried out?

-How did the authors differentiate the results of intracellular replication in J774.1 macrophages and bacterial cytotoxicity in the same cell line?

-In vivo studies: There is no mention of approval of the protocol by the ethics committee involving the use of animals. Different bacterial cell densities were inoculated into tilapia in the in vivo assay. According to the results, at the highest cell density there was no difference between the survival of fish infected with the wild and mutant strains. Why were histological analyzes only performed at this cell density? What does the phrase mean: “The rest of the experiment was repeated three times.”

-2.15. How was the ROS assay designed?

-The manuscript needs revision in the English language.

Minor comments

-The meaning of acronyms/abbreviations must be mentioned in the first description in the text.

-Please use “cell density” instead of “cell concentration”

-If the experiments were performed in replicates and repeated at least three times, the authors must present the results with standard deviations in the images.

Comments on the Quality of English Language

The manuscript needs revision in the English language.

Author Response

We thank the reviewer for his/her suggestions

Comment 1: Some issues should be clarified, mainly related to methodology. A clear description of the methodology for interpreting the results is crucial.

Response:We revised the details of methodology for clear understanding. (Section 2.2, 2.3, 2.7-2.14)

Comment 2: Was the deletion in the DNA-binding domain in the mutant strain confirmed only by PCR?

Response: We used PCR and DNA sequencing of PCR production to confirm the deletion in the DNA-binding domain in the mutant strain. The siderophore production of mutant strain (Fig. 1) further confirmed the mutant.

Comment 3: Why was the TX01∆fur mutant strain included in only a few experiments?

Response:The knockout strain of full-length genes of fur, TX01∆fur, was obtained previously (https://doi.org/10.1016/j.jprot.2016.04.005, reference 22). In a few experiments (Figure 1B and 2A-D), for convenient comparation, we did the same experiments using the TX01, TX01∆fur and TX01∆fur2, but the results of TX01∆fur was similar with the published results. Therefore, the TX01∆fur mutant strain included in only a few experiments.

Comment 4: The standardization of bacterial inocula for experiments must be clearly informed since the authors carry out a comparative analysis between wild-type and mutant strains. For example:  How did the authors ensure that the initial inoculum for growth curve analysis contained the same cell density for the wild-type and mutant strains? According to the methodology, 50 μL of the 16 h cultures were added.

Response:The standardization of bacterial inocula of wild type and mutant strains was vital for comparative analysis. We diluted the overnight cultures of wild type and mutant strains to the similar optical density (OD600) and measured the concentration using the hemocytometer to make sure the same initial inoculum. There is a little vague about the description of the growth curve analysis. We revised the description in the section 2.3.

Comment 5: Transmission electron microscopy: The purpose and results of this methodology are not shown in the main body of the manuscript, only in the supplementary material.

Response:TEM was used to observe the effect of TX01∆fur2 on the change of bacterial flagella. So, we changed the title of 2.8 to display the purpose of methodology. The results were described in the Line 334-336 of revised text. Because there was no difference in the two strains, the figure was attached in the supplementary material.

Comment 6: Does the assay describe bacterial adhesion or invasion into eukaryotic cells?

Response: The assay describes bacterial invasion into eukaryotic cells. We changed the description about the assay for clear understanding. For nonphagocytic cells (Hela and FG cells), we measured the total amount of bacteria adhesion to and invasion into the cells, because we did not kill the extracellular bacteria. For phagocytic cells (RAW264.7 and J774.1 cells), we measured the total amount of bacteria replicated in the cells, because we used the 200 µg/mL gentamicin to kill the extracellular bacteria.

Comment 7: How were the wild-type and mutant strains transformed? Which eukaryotic cells were subjected to the invasion assays? On what equipment were the microscopic analyses carried out?

Response: The competent cells of the wild-type and mutant strains were prepared and transformed with plasmid by electroporation with Gemini Twin Waveform Electroporation systems (Harvard Apparatus BTX, Holliston, MA, USA). The fluorescence microscopy was used to observe the nonphagocytic cells (Hela and FG cells) and phagocytic cells (RAW264.7 and J774.1 cells) infected with TX01 and TX01∆fur2. The microscopic analyses carried out on the confocal laser scanning microscopy (Olympus Fluoview FV1000, Japan). We also revised the description of this assay.

Comment 8: How did the authors differentiate the results of intracellular replication in J774.1 macrophages and bacterial cytotoxicity in the same cell line?

Response: When we did the intracellular replication experiment, we used the DMEM containing 200 µg/mL gentamicin to kill extracellular bacteria. After the TX01 and TX01∆fur2 infected the J774.1 cells for different time, we checked the amounts of intracellular bacteria. For the cytotoxicity caused by bacterial infection, after the J774.1 cells were infected by bacteria (not kill extracellular bacteria), LDH release from the J774.1 cells were checked, which represents its cytotoxicity.

Comment 9: In vivo studies: There is no mention of approval of the protocol by the ethics committee involving the use of animals. Different bacterial cell densities were inoculated into tilapia in the in vivo assay. According to the results, at the highest cell density there was no difference between the survival of fish infected with the wild and mutant strains. Why were histological analyzes only performed at this cell density? What does the phrase mean: “The rest of the experiment was repeated three times.”

Response: we added the approval of the protocol by the ethics committee involving the use of animals in the Line 524-526. The histological analyzes was carried out in the tilapias infected with 107 CFU/fish, not the highest cell density (108CFU/fish). Because there was difference between the survival of fish infected with the wild and mutant strains in this situation. We changed the sentence to “The experiment was repeated three times.”

Comment 10: How was the ROS assay designed?

Response: we added the details of the ROS assay in the Line 258-267 of the revised text. We check the ROS generation of the head kidney macrophages from healthy tilapias with infection of the TX01 or TX01Δfur2

Comment 11: The manuscript needs revision in the English language.

 Response: we revised the English language of text.

Minor comments

Comment 12: The meaning of acronyms/abbreviations must be mentioned in the first description in the text.

Response: We added the meaning of acronyms/abbreviations in the first description in the Line 114, 138, 141, 142, 221,222, 236, 252, 259, and 467of the revised text.

Comment 13: Please use “cell density” instead of “cell concentration”

Response:We changed the “cell concentration” to “cell density” in Line 138, 168, 177, 190, 352, and 354 of the revised text.

Comment 14: If the experiments were performed in replicates and repeated at least three times, the authors must present the results with standard deviations in the images.

Response: All results were presented with standard deviations in the images, except the survival rate (Figure 4). In general, the result of survival rate does not present with standard deviations in the images. We added the sentence “Similar results were observed in other two repeated experiments” in the manuscript (Line 354-355). In addition, the standard deviations are slight in the Figure1A.

Reviewer 2 Report

Comments and Suggestions for Authors

1-     Please substitute the keywords which included in the research title as Edwardsiella piscicida.

2-     In lines: 11 and 162:  the authors should mention different manganese concentrations and different incubation temperatures.

3-     In line 210: please include the system model.

4-     Line 236: pls give a hint about ROS is it in the serum of infected fish or in any other?

5-     Line 391: remove e from wase.

6-     Remove B-actin bands from Fig 7 as this is gel electrophoresis, not PCR result.

7-     Pls, include the title conclusion.             

specific comments:

1-    Regarding the scientific characteristics, the present research is nearly complete work, the methodology covers all required parts, and the conclusion is supported by results.  

2-    There is a novelty point in the present research as it proved that the pathogenicity of E. piscicida is not dependent on bacterial number in host tissues but depends on biofilm production andTX01Δfur2.

3-    This work will be highly valuable for aquaculture community and microbiologist and focusing on important point needs further similar research for other bacterial fish pathogens   

Author Response

We thank the reviewer for his/her suggestions.

Comment 1: Please substitute the keywords which included in the research title as Edwardsiella piscicida.

Response: The keywords were changed to “Fish pathogen; ferric uptake regulator; N-terminal domain; Stress tolerance; Virulence”.

Comment 2: In lines: 11 and 162:  the authors should mention different manganese concentrations and different incubation temperatures.

Response: We revised the details about the manganese resistance and temperature tolerance in the 2.7 section. We used the concentration of 7 mM MnCl2 and high temperature (42 ℃) to determine the bacterial resistance.

Comment 3: In line 210: please include the system model.

Response: we added the system model in the Line 232.

Comment 4: Line 236: pls give a hint about ROS is it in the serum of infected fish or in any other?

Response: we added the details of ROS assay in the Line 258-267. We extracted the head kidney macrophages from healthy tilapias. After the head kidney macrophages infected by the TX01 and TX01Δfur2, the ROS generation of head kidney macrophages was determined.

Comment 5: Line 391: remove e from wase.

Response: we changed the “wase” to “was” in the Line 424.

Comment 6: Remove B-actin bands from Fig 7 as this is gel electrophoresis, not PCR result.

Response: Theβ-actin bands in the Fig 7 was the result of the western blot, which was as the internal reference.

Comment 7: Pls, include the title conclusion.

Response: we added the title conclusion in the Line 506-516.

pecific comments:

1-Regarding the scientific characteristics, the present research is nearly complete work, the methodology covers all required parts, and the conclusion is supported by results. 

2- There is a novelty point in the present research as it proved that the pathogenicity of E. piscicida is not dependent on bacterial number in host tissues but depends on biofilm production andTX01Δfur2.

3-This work will be highly valuable for aquaculture community and microbiologist and focusing on important point needs further similar research for other bacterial fish pathogens 

Reviewer 3 Report

Comments and Suggestions for Authors

The work of Niu et al. is an interesting one which provides new scientific insights of the role of the N-terminal domain of the Fur regulatory protein. Therefore, I recommend the article for publication, however, some revisions must be made. I list below the reasons.

General impression

The Introduction section is adequate for the study. The M&M section needs some revisions. Elements of discussion are found within the Results section. The Discussion section is well written; however, I would like to recommend the separation of the last paragraph as a Conclusion section. The English to my opinion is good enough, thus, some slight polishing is needed.

Major remarks

I am missing the confirmed/rejected hypothesis as a conclusion within the abstract, otherwise the abstract is well written. The PCR reaction mixture and the amplification conditions are missing in section 2.2. 600 nm are not within the UV spectrum (section 2.3)! Please, check the platings in section 2.4 – 10 ul of culture with OD600=1 means that you plated 10^7 cell on the Petri dish…

Minor remarks

Line 2: I would recommend using “Fur protein” instead of only “Fur” in the title

Line 26: “Knocked out”

Line 36: “N-terminal domain”

Lines 257-258, 279-281, 330-334, 346-348, 355-356, 366-368, 396-399: represent discussion

Comments on the Quality of English Language

Despite not being a native English speaker, another round of some slight language polishing is needed.

Author Response

We thank the reviewer for his/her suggestions.

The work of Niu et al. is an interesting one which provides new scientific insights of the role of the N-terminal domain of the Fur regulatory protein. Therefore, I recommend the article for publication, however, some revisions must be made. I list below the reasons.

General impression

Comment 1: The Introduction section is adequate for the study. The M&M section needs some revisions. Elements of discussion are found within the Results section. The Discussion section is well written; however, I would like to recommend the separation of the last paragraph as a Conclusion section. The English to my opinion is good enough, thus, some slight polishing is needed.

Response: We revised the details of material and methods. We think that the elements of discussion found within the Results section help readers understand our results, so hope to keep them. We separated the last paragraph as a Conclusion section. We changed the partial languages of the text.

Major remarks

Comment 2: I am missing the confirmed/rejected hypothesis as a conclusion within the abstract, otherwise the abstract is well written. The PCR reaction mixture and the amplification conditions are missing in section 2.2. 600 nm are not within the UV spectrum (section 2.3)! Please, check the platings in section 2.4 – 10 ul of culture with OD600=1 means that you plated 10^7 cell on the Petri dish…

Response: We fail the meaning of “I am missing the confirmed/rejected hypothesis as a conclusion within the abstract, otherwise the abstract is well written”. In the section 2.2, the PCR reaction mixture and the amplification conditions are added. In the section 2.3, we used the ultraviolet and visible (UV-Vis) spectrophotometer to determine the OD600 of bacteria and we corrected the name of equipment. In the section 2.4, we really dropped the 10 μL (about the 10^5 cell) of E. piscicida (OD600=1, about 10^7 CFU/ml) on the plate and keep the bacterial drop motionless, not plating the bacterial drop on the plate.

Minor remarks

Comment 3: Line 2: I would recommend using “Fur protein” instead of only “Fur” in the title

Response: we changed the “Fur” to “Fur protein” in the title.

Comment 4: Line 26: “Knocked out”

Response: we changed the “knockout” to “knocked out” in the Line 26.

Comment 5: Line 36: “N-terminal domain”

Response: we changed the “N-terminal” to “N-terminal domain” in the Line 36.

Comment 6: Lines 257-258, 279-281, 330-334, 346-348, 355-356, 366-368, 396-399: represent discussion

Response: We think these descriptions help readers understand our results, so hope to keep them.

Comment 7: Despite not being a native English speaker, another round of some slight language polishing is needed.

Response: we revised the English language of text.

Round 2

Reviewer 1 Report

Comments and Suggestions for Authors

Thank you for all replies.

Comments on the Quality of English Language

Minor editing of English language is required.